# Is the IL1RA/IL1B Ratio a Suitable Biomarker for Subclinical Endometritis in Dairy Cows?

**DOI:** 10.3390/ani12233363

**Published:** 2022-11-30

**Authors:** Anna Maria Kneidl, Christina Deborah Marth, Sandra Kirsch, Frank Weber, Yury Zablotski, Anika Luzia Helfrich, Simone Tamara Schabmeyer, Julia Katharina Schneider, Wolfram Petzl, Holm Zerbe, Marie Margarete Meyerholz-Wohllebe

**Affiliations:** 1Clinic for Ruminants with Ambulatory and Herd Health Services, Centre for Clinical Veterinary Medicine, Ludwig Maximilian University Munich, 85764 Oberschleissheim, Germany; 2Melbourne Veterinary School, Faculty of Veterinary and Agricultural Sciences, The University of Melbourne, Werribee, VIC 3030, Australia

**Keywords:** endometrium, chronic inflammation, bovine interleukin 1-family, pro- and anti-inflammatory cytokines, regulation of inflammation, infertility

## Abstract

**Simple Summary:**

Chronic inflammation in the uterus after calving leads to reduced fertility in dairy cows, resulting in huge economic losses for dairy farmers as well as the premature removal of impacted animals. Correctly identifying and diagnosing affected cows allows for the appropriate and timely treatment of the condition. This study aimed to compare the concentrations of two factors associated with the early immune response in secretions collected from the uterus of cows with and without chronic uterine inflammation. One of these factors is known to cause (further) inflammation while its counterpart generally reduces inflammation in tissues. We found that the factor increasing inflammation was present in significantly higher concentrations in the uterus of cows diagnosed with prolonged uterine inflammation compared with cows without the inflammation. The factor reducing inflammation was present in similar concentrations in both groups of cows, but the ratio of the two measured proteins also differed between healthy and diseased cows. In conclusion, this study highlights the importance of the two measured immune factors in the development of inflammation in the uterus, as well as their potential as a diagnostic tool.

**Abstract:**

The adequate expression of cytokines is essential for the prevention and healing of bovine endometrial inflammation. This study investigated the intra-uterine concentration of the proinflammatory cytokine interleukin (IL)1B and its antagonist IL1RA in cows with and without subclinical endometritis (SE). Samples were taken from 37 uteri at the abattoir and 26 uteri in vivo. Uterine secretion samples were classified as showing no signs of SE (SEneg; polymorphonuclear neutrophil granulocyte (PMN) < 5%) or showing signs of SE (SEpos; PMN ≥ 5%). Concentrations and ratios for IL1B and IL1RA were measured using a commercial and a newly established AlphaLISA kit, respectively. In both groups, a higher concentration of IL1B was detected in the SEpos group compared with the SEneg group (abattoir: *p* = 0.027; in vivo *p* < 0.001). No significant differences were observed in the concentration of IL1RA (*p* > 0.05). In uterine secretion samples retrieved in vivo, a lower IL1RA/IL1B ratio was detected in the SEpos group compared with the SEneg group (*p* = 0.002). The results of this study highlight the important role of IL1B and IL1RA during endometritis and the potential of the IL1RA/IL1B ratio as a possible biomarker for SE.

## 1. Introduction

Uterine inflammation causes economic losses to the dairy industry worldwide due to reduced fertility and animal welfare of affected cows. Prevention and therapy of subclinical endometritis (SE) represents a challenge for owners and veterinarians, as cows remain clinically healthy with no outward signs of disease. The economic impact of SE results from lower pregnancy rates, longer calving to conception intervals and higher culling rates due to subfertility [1,2].

In healthy cows, potential pathogens are eliminated, and the initial non-inflammatory condition of the endometrium is reestablished by day 30 after calving [3]. Neither macroscopic nor microscopic alterations of the uterus should be present and regular ovarian activity should be observable. Endometritis is defined as ongoing uterine inflammation later than 21 days after parturition and is divided into clinical and subclinical endometritis [4]. Clinical endometritis (CE) in cattle is characterized by purulent or mucopurulent vaginal discharge without systemic symptoms [4,5]. Subclinical endometritis (SE) is defined as clinically unapparent inflammation of the endometrium [4,6,7]. SE is usually diagnosed by cytological samples that can be obtained via cytobrush swab samples or low-volume lavage [6,7,8]. A threshold of 5% polymorphonuclear cells (PMN) in cytobrush samples taken between 21 and 62 days after calving has been shown to be a reliable diagnostic marker for SE [9,10,11]. In contrast to equine medicine, both diagnostic sample collection procedures are usually limited to scientific approaches in cattle and are not commonly used in routine veterinary care at the moment [7].

The pathogenesis of SE is still not fully understood. Several studies have established that bacterial infections alone do not cause SE [12,13]. However, there is some debate regarding the cause of the chronic nature of SE. Insufficient down-regulation of the immune system may lead to an excessive and persistent immune response [14]. Alternatively, an inefficient initial immune response may impair complete elimination of bacterial infections resulting in a persistent inflammation [15]. An ongoing inflammatory response after successful elimination of bacterial stressors is another possible pathway to SE [12].

All hypotheses agree that the adequate regulation of pro- and anti-inflammatory factors, such as interleukins (IL), is crucial to reestablish the uterine homeostasis without inflammation. Proinflammatory cytokines like IL1B have been reported to be upregulated in uterine cytological, tissue and secretion samples of cows with SE [14,16,17]. The local finetuning of pro- and anti-inflammatory cytokines to overcome dysregulation during SE is complex. In this context, IL1 receptor antagonist (IL1RA) is a promising target to further investigate the local immune dysregulation during SE. When IL1RA binds to interleukin receptor 1, it makes the receptor unavailable for the proinflammatory action of IL1A and IL1B [18]. To efficiently reduce inflammation, the IL1RA concentration must be 10 to 1000-fold higher than that of the agonists [19] making the ratio of IL1RA and IL1B an important tool to assess the dysregulation of the immune response during SE.

Amplified luminescent proximity homogeneous assay-linked immunosorbent assay (AlphaLISA) are proximity-based assays using donor beads that can be excited by light at a wavelength of 680 nm and emit a luminescent signal at 615 nm if in close proximity to an acceptor bead conjugated to a specific antibody. The signal strength of the emitted signal is proportional to the concentration of target protein in a sample [20]. The main advantage of this type of assay is that it can accurately measure the concentration of proteins in very small sample volumes with high sensitivity [21]. In clinically healthy uteri, only small volumes of uterine secretions (US) are present, making this assay particularly suitable for the assessment of their composition. An AlphaLISA is already commercially available for bovine IL1B and has previously been confirmed as suitable for detection of the cytokine in US [17].

Therefore, the aims of this study were: (a) to establish and validate an AlphaLISA for IL1RA, and (b) to investigate the suitability of the IL1RA/IL1B ratio as a potential biomarker for SE in cows sampled at the abattoir and in vivo.

## 2. Materials and Methods

### 2.1. Collection of Uterine Secretions and Assessment of Dilution Factors and Blood Contamination

Samples were taken from the uterus of 37 cows of different breeds at the abattoir. Breeds included Holstein-Friesian, Red Holstein, Brown Swiss, Simmental and beef breeds. The average age of cows in this group was 7 years (range: 3.8–8.7 years). Only uteri that had not been contaminated during the process of slaughtering were used for sample collection. All uteri were intact, symmetrical and did not show any pathological alterations. This was used to assume completion of uterine involution and no signs of early pregnancy. The attached ovaries contained at least one corpus luteum (CL) and displayed no signs of pathological alterations.

Additionally, IL1B and IL1RA concentrations were measured in US samples collected from 26 systemically and genitally healthy cows kept at the Bavarian State Research Centre for Agriculture. These cows were Simmental, Brown Swiss, Red Holstein or cross-breeds with an average of 2.7 lactations (range: 1–6 lactations). Fertility data for these cows have been outlined previously [17]. In the in vivo sampling group, samples were collected from cows between days 45 and 60 after parturition and cycle stage was evaluated by transrectal palpation and ultrasonography using a 10 MHz linear array transducer (Esaote, Cologne, Germany).

All samples were collected using the Munich multi-functional sample collection tool (MMSCT). This tool was developed in our research group and has previously been described for sample collection in uteri from the abattoir [22] and in vivo [17]. Briefly, the tool consists of a working channel through which a Merocel sponge (Medtronic; Dublin, Ireland) and a cytobrush sample can be collected with just one passage through the cervix. In the present study, the Merocel sponge was advanced through the sterile working channel and left for 4 min in uteri at the abattoir and 2 min in uteri of cows in the in vivo group. After removal from the uterus, the sponges were immediately placed into a micro tube containing 300 μL phosphate-buffered saline (PBS) and 100 μg/mL aprotinin (extraction buffer). Tubes were then weighed using a precision scale to assess their dilution factor, as previously described [17]. Next, sponges were centrifuged at 14,000× *g* and 20 °C for 15 min in filtration tubes. The elution and centrifugation steps were repeated before the combined eluates were assessed for blood contamination using optical density measurements at 570 nm as previously described [17]. Subsequently, the US eluate was stored at −80 °C until further processing.

### 2.2. Bacteriological and Cytological Evaluation of Uterus Samples and Group Allocation

The uteri sampled at the abattoir were incised and a swab sample was collected for bacteriological evaluation on sheep blood agar, violet red bile agar and Edwards agar. In addition, a cytobrush sample was collected and immediately rolled onto a glass slide for cytological evaluation. In the in vivo group, bacteriological and cytological sampling were performed via the MMSCT as previously described [17]. The cytobrush sample was streaked onto two sterile glass slides for cytological evaluation before being rolled onto Columbia sheep blood agar for bacteriological evaluation.

For both sampling groups, an additional swab sample was taken from the filter of the microtube after the Merocel sponge filtration and streaked on Columbia sheep blood agar only for the in vivo and sheep blood agar, violet red bile agar and Edwards agar for the abattoir group. All bacteriological samples were incubated for 48 h at 37 °C. Bacteriological growth was evaluated after 24 h and 48 h and bacteriological identification was performed as described previously [17]. Briefly, plates with ≥5 colony-forming units of ≤2 different colony morphologies were regarded as bacteriologically positive. Bacterial species were differentiated using matrix-assisted laser desorption mass spectrometry (Microflex LT and MALDI Biotyper 2.0, Bruker, Billerica, MA, USA).

For cytological evaluation, glass slides were air-dried, stained with Diff-Quick and evaluated by light microscopy (×400 magnification). Cytological classification was based on the percentage of PMN out of 300 nucleated cells. Based on cytological examination, the uterine secretion samples were divided into the following two groups: Those showing no sign of subclinical endometritis (SEneg; PMN < 5%) and those from animals with subclinical endometritis (SEpos; PMN ≥ 5%). This threshold was based on previous studies as outlined in the introduction [9,10,11].

### 2.3. Establishment and Evaluation of Bovine IL1RA AlphaLISA

To quantify protein concentrations of IL1RA in bovine uterine secretions, an IL1RA AlphaLISA was established and validated with unconjugated AlphaLISA acceptor beads, streptavidin conjugated AlphaLISA donor beads (PerkinElmer LAS, Rodgau, Germany) and the specific antibody in purified and biotinylated form (Bio-Rad, Hercules, CA, USA). Several concentrations of reagents and incubation times were assessed to identify the combination resulting in the most accurate measurement of IL1RA concentrations. In the final protocol, 5 μL of sample or standard was mixed with 2.5 μL anti-IL1RA conjugated acceptor beads (200 μg/mL) and 2.5 μL biotinylated antibody (200 nM) on a 384-well alpha plate and incubated for 2 h at room temperature (21 °C). Next, 40 μL Streptavidin donor beads (50 μg/mL) was added to each well followed by another 30 min incubation step at room temperature under light protection (<100 Lux). The photometric measurement was performed at 615 nm using a CLARIOstar microplate reader (BMG Labtech, Ortenberg, Germany) with a 384-well format focal lens equipped with an AlphaLISA optical module. Serial dilutions of standard up to a concentration of 975 pg/mL were included on each plate.

To determine the reliability of the established assay, several assay parameters were defined. The lower detection limit (LDL) was determined by replacing the sample volume with AlphaLISA HiBlock Buffer (Perkin Elmer, Waltham, MA, USA). In combination with the standard curve, this measure determines the minimum IL1RA concentration at which the signal could reliably be distinguished from background noise. The intra-assay coefficient of variation (CV) was determined by running each standard in triplicate on each plate, calculating the CV for each well and averaging the results across 18 runs. The inter-assay CV was determined by analyzing the US of three randomly selected cows on three separate plates analyzed over three days, each time run in triplicate. The hook point, at which an increase in concentration of target protein starts to inhibit the assay, was evaluated using standard dilutions up to 2000 ng/mL. The recovery rate was established by spiking wells containing AlphaLISA HiBlock Buffer or US of predetermined IL1RA concentration with purified recombinant bovine IL1RA to evaluate how much of the protein could be recovered.

### 2.4. Detection of IL1B, Correction for Blood Contamination and Calculation of IL1RA/IL1B Ratio in Bovine Uterine Secretions

To measure IL1B concentration in the uterine secretions, a commercially available AlphaLISA was used (Perkin Elmer, USA) according to the manufacturer’s instructions. The IL1B AlphaLISAs were performed using the concentrations and times described in 2.3. Each negative control was run in six replicates, each standard in triplicate and each sample in duplicate. If the intra-assay CV was >10%, the assay was repeated.

If blood contamination of >2.18% was detected in a sample, the IL1B and IL1RA concentrations were corrected based on protein concentrations in a paired blood plasma sample collected from the tail vein of the cow, as previously described [17]. If blood contamination >33% was detected, the sample was excluded from all further analyses.

To calculate the IL1RA/IL1B ratio, IL1RA concentrations were divided by IL1B concentrations for each animal. The sample collection and analysis protocols have been summarized in Figure 1.

### 2.5. Statistical Analysis and Graphical Illustration

Statistical analyses were performed using R software for statistical computing (Version 4.1.1) [23]. Results of descriptive statistics were presented as mean and SD for normally distributed and median for non-normally distributed data. Initially, pairwise comparisons were performed to detect differences in log-transformed data of IL1B and IL1RA concentrations as well as the IL1RA/IL1B ratio between SEpos and SEneg cows within each sampling group (abattoir and in vivo). In a second step, data were analyzed for differences between the sampling groups within each uterine health groups (SEpos and SEneg cows). To do this, data were first tested for normal distribution via Shapiro–Wilk test and samples were compared for homogeneity of the variances via Levene test. Variances were distributed homogenously in all cases; thus, the Student’s *t*-test was used for pairwise comparisons of normally distributed data and the Mann–Whitney U-test for non-normally distributed data. No *p*-value correction for multiple comparisons was applied due to the exploratory character of our study.

Differences with *p* < 0.05 were considered as statistically significant. Calculated differences with *p* < 0.1 were considered as statistical tendencies.

## 3. Results

### 3.1. Cycle Stage, Bacteriological and Cytological Evaluation of Donor Cow Uteri

In the abattoir group, all sampled cows were in dioestrus. In the in vivo group, 15 cows were in dioestrus (11 classified as SEneg and 4 classified as SEpos), and 11 cows were in oestrus (10 classified as SEneg and 1 classified as SEpos).

In the abattoir group, bacteria were cultured from five uteri. The detected bacteria were characterized as *Escherichia coli* (*E. coli*, *n* = 2), *Trueperella pyogenes* (*n* = 1), *Enterococcus faecium* (*n* = 1) and *Actinobacillus seminis* (*n* = 1). In the in vivo group, bacteria were cultured from the US of one animal (mix of *Streptococcus uberis* and *E. coli*).

In the abattoir group, the PMN percentage in the cytobrush samples ranged from 0% to 43.9% (median 0.1%). Three animals in the abattoir group were diagnosed with SE based on a PMN percentage ≥ 5% (SEpos) and 34 animals showed no sign of SE (SEneg). In the in vivo group, the PMN percentage in the cytobrush samples ranged from 0% to 26.8% (median 0.7%). Five animals in the in vivo group were categorized in the SEpos group and twenty-one in the SEneg group based on the same parameters as the abattoir group.

Due to the small number of bacteriologically positive samples in each group and the standardized classification of SE, all further analyses focused only on PMN proportion to discuss differences between SEpos and SEneg animals.

### 3.2. Dilution Factors and Blood Contamination of Uterine Secretion Samples

The individual dilution factors for US ranged from 1.4 to 32.1 (median: 4.5) in the abattoir group (*n* = 37) and from 2.7 to 41.0 (median: 9.0) in the in vivo group (*n* = 26). The blood contamination of the US in the abattoir group ranged from 0.2% to 2.7% (median 1.0%) and from 2.2% to 22.4% (median 3.9%) in the in vivo group.

### 3.3. Quality Parameters Confirm Reliability of Established Bovine IL1RA AlphaLISA

An approximately linear course was detected for the IL1RA AlphaLISA between 31.25 ng/mL and 250 ng/mL IL1RA. The lower detection limit (LDL) was 15.750 relative fluorescence units, which is equivalent to 3.5 ng/mL. The intra-assay CV was <10% for 97.6% of the included reading points and only 1.6% showed an intra-assay CV of >15%. The mean intra-assay CV was 4.9% (SD = 2.9%). The inter-assay CV was 4–7%. The hook point was detected starting at a concentration of 500 ng/mL IL1RA. The recovery rate of samples with IL1RA concentrations >5 ng/mL was predominantly found between 75% and 125%. A summary of quality parameters for the assay can be found in Table 1 below. By establishing a reliable AlphaLISA for the detection of bovine IL1RA, a suitable method to quantify IL1RA concentrations in US of cows was developed within this study.

### 3.4. Detection of IL1B, IL1RA and IL1RA/IL1B Ratio in Bovine Uterine Secretions

All results were assessed for blood contamination, as previously described [24]. Protein concentrations in US were corrected according to protein concentrations detected in paired plasma samples. Only one animal showed detectable IL1RA concentrations in blood plasma (42.9 ng/mL) and three animals had detectable IL1B concentrations in blood plasma, of which two were below the lower detection limit of 2.26 ng/mL and were corrected based on the lowest standard curve point. The remaining animal’s plasma contained 73.2 pg/mL IL1B.

#### 3.4.1. Lower IL1RA/IL1B Ratio in Cows with SE

In the abattoir group, IL1B and IL1RA concentrations in uterine secretions were higher in samples of cows in the SEpos compared with the SEneg group (IL1B: *p* = 0.027; IL1RA: *p* = 0.071). In samples collected from the abattoir, the IL1RA/IL1B ratio tended to be lower in US in the SEpos compared with the SEneg group (*p* = 0.082, Figure 2).

In the in vivo group, IL1B was significantly higher in US of cows in the SEpos compared with the SEneg group (*p* < 0.001). The IL1RA concentration did not differ between SEpos and SEneg cows (*p* > 0.1). The IL1RA/IL1B ratio in US was significantly lower in SEpos compared with SEneg cows (*p* = 0.002; Figure 2).

#### 3.4.2. Lower IL1RA and IL1RA/IL1B Ratio in SEneg Cows Sampled at the Abattoir

Within samples in the SEneg group, no significant differences were detected in the IL1B concentration of US from cows sampled at the abattoir or in vivo (*p* > 0.1). The IL1RA concentration tended to be lower in US of cows sampled at the abattoir compared with cows sampled in vivo (*p* = 0.070). The IL1RA/IL1B ratio was significantly lower in US of cows sampled at the abattoir compared with cows sampled in vivo (*p* = 0.019; Figure 2).

#### 3.4.3. No Differences of IL1B, IL1RA and IL1RA/IL1B Ratio in SEpos Cows Independent of Sampling Type

Within samples in the SEpos group, no significant differences were detected in either the IL1B, ILRA concentration or the IL1RA/IL1B ratio in US of cows sampled at the abattoir or in vivo (*p* > 0.1; Figure 2).

## 4. Discussion

### 4.1. SE Positive Cows Show a Proinflammatory Shift in the IL1RA/IL1B Ratio in Uterine Secretions

Uterine secretions have a different composition in cows with SE compared with those with uninflamed uteri [17,22,25]. The cytokine IL1B has previously been shown to be expressed in higher concentrations in uterine cytological, tissue and secretion samples of cows with SE compared with cows without SE [14,16,17]. This suggests a proinflammatory uterine environment. In healthy cows, the proinflammatory effect of IL1B should be adequately downregulated via its anti-inflammatory counterpart IL1RA. Therefore, we hypothesized that IL1RA also changes in US according to the uterine health status of the cow and that the IL1RA/IL1B ratio represents a potential biomarker for SE in cows in vivo and in uteri collected at the abattoir.

To test this hypothesis, an IL1RA AlphaLISA was successfully established. The evaluation of the assay via standard quality indicators confirmed the suitability and reliability of the newly established IL1RA assay. Therefore, it was used to quantify IL1RA concentrations in US of cows.

In both the abattoir and the in vivo group, the IL1B concentration in US was higher in SEpos cows (PMN ≥ 5%) compared with cows with healthy uteri. In contrast, only a trend for higher IL1RA concentrations in SEpos cows could be detected in the abattoir group. In the in vivo group, only numerically higher IL1RA concentrations were measured in SEpos cows. Similar results were obtained in a study by Gabler et al. [26] in which no differences in IL1RA gene expression were detected in endometrial epithelial cells obtained via cytobrush from cows with and without SE.

In the present study, the IL1RA concentration clearly exceeded the IL1B concentration, irrespective of the sampling group or the presence of SE. In both sampling groups, US showed a lower IL1RA/IL1B ratio in SEpos compared with SEneg cows, but this result was more prominent in cows sampled in vivo compared with cows sampled at the abattoir. The observed shift of the IL1RA/IL1B ratio towards a proinflammatory state in SE cows seems to be driven by the higher IL1B concentration. The question whether this shift is the reason for or the consequence of the inflammatory environment in US of cows with SE cannot be answered with the present dataset. The shift could reflect a dysregulation of the local endometrial immune mechanisms with overexpression of pro- in relation to anti-inflammatory factors. Alternatively, the shift in the IL1RA/IL1B ratio could be the consequence of the subclinical inflammation in the uterus. The SE itself might originate from a chronic bacterial or viral infection of the endometrium that persists in the uterus due to its inefficient cleaning mechanisms [27]. In both cases, fertility is likely impacted by the IL1RA/IL1B shift in US, as adequate regulation of the IL1-system is necessary for the success of reproductive events, such as implantation [28,29].

Due to the low sample numbers in the SEpos groups (*n* = 3 sampled at the abattoir and *n* = 5 sampled in vivo), the results must be seen as an exploratory study and should not be overinterpreted. Confirmation of the observed results in future studies comprising of a larger sample size would be beneficial. It would be of great interest to include holistic analyses to collectively investigate the uterine transcriptome, proteome, and microbiome for the identification of further potential biomarkers.

### 4.2. Uterine Secretions of SEpos Cows Obtained at the Abattoir Reflect the Situation In Vivo

According to the 3R principle, it is desirable to develop alternatives to animal experiments wherever possible. Abattoir samples can be quickly and easily accessed without impacting animal welfare, but slaughter may impact gene or protein expression in tissues. In the present study, US were obtained at the abattoir and in vivo. The two sampling methods were compared regarding sample volume, individual dilution factors, blood contamination and protein concentration of IL1B and IL1RA.

In the abattoir group, a higher volume of US was collected compared with US obtained in vivo. A possible explanation for this discrepancy is the differing duration in which the Merocel Sponge remained in the uterus (4 min in uteri at the abattoir and only 2 min in uteri of cows sampled in vivo). The shorter duration in vivo was chosen due to animal welfare concerns. However, an increase in intrauterine fluid accumulation after death cannot be ruled out. The impact of changes after death were sought to be minimized by collecting samples within 60 min after slaughter.

To evaluate whether death or slaughter influence the protein concentration of IL1B, IL1RA or the IL1RA/IL1B ratio, the results were compared between samples collected at the abattoir and in vivo. In samples from SEneg cows, the IL1RA/IL1B ratio in US was lower in the abattoir group compared with SEneg cows sampled in vivo. One possible explanation for this discrepancy is the very heterogeneous nature of the SEneg group at the abattoir, particularly in terms of sample time in relation to parturition. In contrast, US from cows sampled in vivo were all collected between 45 and 60 days after parturition and from cows housed under the same conditions. This may have led to the wider range of IL1B and IL1RA levels in the group sampled at the abattoir which only reached significance when combined into the IL1RA/IL1B ratio. In contrast, no significant differences were detected between the two sampling methods in SEpos cows in which the apparent inflammation likely outweighed the variations caused by sampling time.

While the IL1RA/IL1B ratio differed significantly between SEpos and SEneg cows within the in vivo group, in the abattoir only a trend could be detected. This discrepancy may be explained by the change in tissue oxygen supply due to exsanguination resulting in a changed inflammatory environment in all samples collected after death. Correlations between ischemia or hypoxia and increased serum concentrations of proinflammatory cytokines have been described previously [30,31].

Interestingly, the results were comparable between sampling methods in the SEpos group. This indicates that the inflammation caused by SE results in cytokine changes in US even in cows sampled at the abattoir. In future studies, this allows for easier access to larger sample numbers which can then be used in combination with explant culture challenge experiments and proteomic analyses to investigate the pathogenesis of SE in more detail.

### 4.3. Is the IL1RA/IL1B Ratio a Suitable Biomarker for Subclinical Endometritis in Dairy Cows?

In the present study, the observed shift of the IL1RA/IL1B ratio towards a proinflammatory state seems to be primarily driven by higher IL1B concentrations. This might initially suggest that IL1B provides the same information as the IL1RA/IL1B ratio regarding the inflammatory condition in the uterus. However, the IL1RA/IL1B ratio also assesses the missing downregulation of inflammatory conditions during SE. High IL1RA and relatively low IL1B expression in a high ratio may indicate a sufficiently regulated inflammation, while low IL1RA and high IL1B values lead to a low ratio indicating ongoing processes of inflammation. Therefore, the Il1RA/IL1B ratio provides a more detailed insight into the regulatory processes of inflammation.

Clearly, this study only indicates the potential of the IL1RA/IL1B ratio, and the robustness of this parameter must be evaluated in further studies with higher sample numbers.

## 5. Conclusions

An AlphaLISA for the quantification of bovine IL1RA concentrations in US of cows was successfully developed within this study. Our results highlight the importance of IL1B and IL1RA protein expression in the context of bovine uterine inflammation. The results also suggest the IL1RA/IL1B ratio as a possible biomarker for an inflamed endometrium which may help to assess the missing downregulation of inflammatory conditions during SE. Whether the observed shift in the IL1RA/IL1B ratio reflects a cause for or a consequence of SE needs to be examined in further studies. Regardless, this information may make it possible to identify ongoing latent endometrial inflammation in cows. This would improve diagnostic, preventative and therapeutic strategies for cows with SE.

## Figures and Tables

**Figure 1 animals-12-03363-f001:**
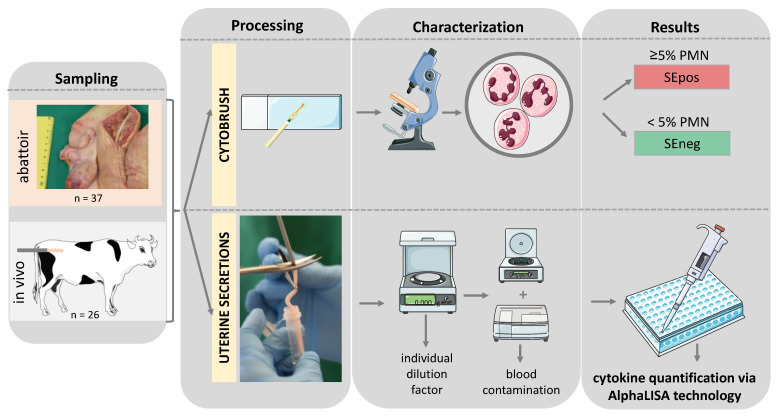
Schematic illustration of the experimental setup of this study. PMN—polymorphonuclear neutrophil granulocyte; SEpos/SEneg—positive/negative for subclinical endometritis (SE) detected via cytobrush; AlphaLISA—amplified luminescent proximity homogeneous assay-linked immunosorbent assay. Graphical elements retrieved from Servier medical art.

**Figure 2 animals-12-03363-f002:**
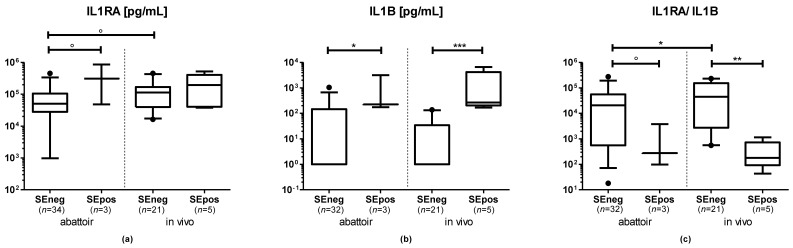
Concentrations of (**a**) interleukin 1 receptor antagonist (IL1RA), (**b**) interleukin 1 B (IL1B) and (**c**) the IL1RA/IL1B ratio in uterine secretions from cows sampled at the abattoir or in vivo measured via amplified luminescent proximity homogeneous assay-linked immunosorbent assay (AlphaLISA). Visualization is based on untransformed data. Boxes extend from the 25th to the 75th percentile, whiskers extend from the 5th to the 95th percentile, values beyond these upper and lower bounds are marked with black dots. The symbols indicate the results of pairwise comparisons of log-transformed data as follows: ° = *p* < 0.1; * = *p* < 0.05; ** = *p* < 0.01; *** = *p* < 0.001.

**Table 1 animals-12-03363-t001:** Quality parameters for bovine interleukin receptor antagonist (IL1RA) AlphaLISA.

Quality Parameter	Value
Standard curve range	31.25–250 ng/mL
Lower Detection Limit	3.5 ng/mL
Intra-Assay Coefficient of Variation	<10%
Inter-Assay Coefficient of Variation	≤7%
Hook Point	500 ng/mL
Recovery rate	75–125%

## Data Availability

Data are contained within the article. Raw data are available on request from the corresponding author.

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
