# Peer review of "Is the IL1RA/IL1B Ratio a Suitable Biomarker for Subclinical Endometritis in Dairy Cows?"

_animals, 2022, doi:10.3390/ani12233363_

Round 1

Reviewer 1 Report

This study established the quantification method of bovine IL1RA using the AlphaLISA assay, and found a difference in IL1RA/IL1B ratio between the healthy cows and the cows with subclinical endometritis. They suggested that the IL1RA/IL1B could be a potential marker of inflamed endometrium. The study is well designed with clear presentation and fluent English writing. However, a confirmation study is required to include larger sample number the finding.

Author Response

Thank you for your comments, we have addressed them below. Please note that all line numbers refer to the manuscript with track changes.

Reviewer comment 1: 

This study established the quantification method of bovine IL1RA using the AlphaLISA assay, and found a difference in IL1RA/IL1B ratio between the healthy cows and the cows with subclinical endometritis. They suggested that the IL1RA/IL1B could be a potential marker of inflamed endometrium. The study is well designed with clear presentation and fluent English writing. However, a confirmation study is required to include larger sample number the finding.

Response 1:

Due to the nature of this exploratory study, we will not be able to include more animals at this stage. We have addressed this limitation in the discussion (l.337-342). We completely agree with the reviewer that our results need to be confirmed by a larger sample size in future studies and have outlined this in the discussion (l.393-395).

Reviewer 2 Report

This is a good study design. However there no unique results. IL-1b confirmed that IL-1b is known to be highly secreted in endometritis cows. For example Foley et al, 2015 report expression of  cytokines and TL receptors.   For the microbiology study,  the authors should exclude since the identification of bacteria was not detailed, and is based on classical microbiological identification which fails to detect most of non-cultured bacteria. In addition, the authors do not discuss the bacteriology results.  The authors should exclude bacteriology  since it does not complement the aim of the study - which is based on measurement of IL-1b and its antagonistic receptor. 

Author Response

Thank you for your comments, we have addressed them below. Please note that all line numbers refer to the manuscript with track changes.

Rewiewer comment 1:

This is a good study design. However there no unique results. IL-1b confirmed that IL-1b is known to be highly secreted in endometritis cows. For example, Foley et al, 2015 report expression of cytokines and TL receptors.   

Response to comment 1:

To the best of our knowledge, this is the first time protein levels of IL1B and IL1RA have been reported in uterine secretions. Foley et al. (2015) have reported IL1B protein levels in blood plasma. While there may be some correlation between protein levels in plasma and uterine secretions, they are unlikely to be identical. In the uterus, IL1RA expression has only been reported on the mRNA level in cows after parturition.

Reviewer comment 2:

For the microbiology study, the authors should exclude since the identification of bacteria was not detailed, and is based on classical microbiological identification which fails to detect most of non-cultured bacteria. In addition, the authors do not discuss the bacteriology results.  The authors should exclude bacteriology since it does not complement the aim of the study - which is based on measurement of IL-1b and its antagonistic receptor. 

Response to comment 2:

Culture-based bacteriological examination of samples is part of the standard reporting for animals with subclinical endometritis. We recognise the value of sequencing for deeper bacteriological evaluation but, as the reviewer rightly suggests, this was not the main focus of the present study. We have included more details regarding the methods used to identify bacteria (l. 152 ff.). We have also added a sentence in the results section, clarifying that the results of the bacteriological evaluation were not used for other analyses in this manuscript (l. 241 ff.).

Reviewer 3 Report

In this article, the author investigated the intra-uterine concentration of the proinflammatory cytokine interleukin (IL)1B and its antagonist IL1RA in cows with and without subclinical endometritis (SE). Samples were taken from 37 uteri at the abattoir and 26 uteri in vivo. Concentrations and ratios for IL1B and IL1RA were measured. Finally, the author highlights the importance of the two measured immune factors in the development of uterus inflammation but also their potential as a diagnostic tool. However, the manuscript's readability could be much improved to better convey the importance of your study. With editing and some minor revisions, I feel that this manuscript will be suitable for publication.

1.      It is better not to have more than 6 keywords in an article.

2. P value should be capitalized and italicized in statistical analysis, please check the entire article for consistent formatting.

3. Line 37, the “in vivo” should be italic.

4. % and ℃ can be combined with numbers without blank spaces. Please check the entire article for consistent formatting.

5. In this manuscript, many units are not written in a standardized way. For example, "μl", "ng/ml", "μg/ml", "pg/ml", including but not only these. Please use SI units.

6. A total of 28 references were cited for this article. Fifteen references were from before 2010, of which 5 were from before 2000. For innovative articles, references should be nearly 3-5 years. Please update references.

7In this article, the grammar should be revised.

Author Response

Please see attachment. Note that all line numbers refer to the manuscript with track changes.

Reviewer 4 Report

The authors of this manuscript follow the intra-uterine concentration of the proinflammatory cytokine interleukin (IL)1B and its antagonist IL1RA in cows with and without subclinical endometritis.

The article could be published, with some recommendations attached.

Basically it refers to:

insertion of references, where requested.

Clarification of abbreviations used

Language review, some specific words are spelled incorrectly

Explanation of the composition of the groups and the required completions.

Description of the types of uterine microbiota and the frequency of isolation.

The conclusion must be rewritten.

Author Response

Please see the attachment. Note that all line numbers refer to the manuscript with track changes.

Round 2

Reviewer 2 Report

Review comments were adequately addressed.